# A nuclease- and bisulfite-based strategy captures strand-specific R-loops genome-wide

**Phillip Wulfridge[1,2], Kavitha Sarma[1,2]***

[1]Gene Expression and Regulation program, The Wistar Institute, Philadelphia, United States; [2]Epigenetics Institute, University of Pennsylvania, Philadelphia, United States

**Abstract** R-loops are three-stranded nucleic acid structures with essential roles in many nuclear processes. However, their unchecked accumulation is associated with genome instability and is observed in neurodevelopmental diseases and cancers. Genome-wide profiling of R-loops in normal and diseased cells can help identify locations of pathogenic R-loops and advance efforts to attenuate them. We present an antibody-independent R-loop detection strategy, BisMapR, that combines nuclease-based R-loop isolation with non-denaturing bisulfite chemistry to produce genome-wide profiles that retain strand information. BisMapR achieves greater resolution and is faster than existing strand-specific R-loop profiling strategies. In mouse embryonic stem cells, we apply BisMapR to find that gene promoters form R-loops in both directions and uncover a subset of active enhancers that, despite being bidirectionally transcribed, form R-loops exclusively on one strand. BisMapR reveals a previously unnoticed feature of active enhancers and provides a tool to systematically examine their mechanisms in gene expression.

## Introduction

R-loops are three-stranded nucleic acid structures that frequently occur during transcription when newly transcribed RNA base pairs with the DNA template strand, forming a DNA:RNA hybrid (*Thomas et al., 1976*). The non-template strand is then extruded as single-stranded DNA (ssDNA). R-loops play important roles in many nuclear processes including recombination, transcription termination, and DNA repair (*Chédin, 2016*; *Niehrs and Luke, 2020*). R-loop formation in mitosis ensures faithful chromosome segregation (*Kabeche et al., 2018*). While R-loops at some genomic sites clearly have beneficial roles, their aberrant accumulation at others is associated with genomic instability and disease (*Crossley et al., 2019*; *García-Muse and Aguilera, 2019*; *Perego et al., 2019*; *Richard and Manley, 2017*). The evident role of R-loops in both cellular function and disease makes it critical that reliable methods exist for profiling their formation across the genome. These would allow for detection of changes in R-loop levels between conditions and for the characterization of features associated with their formation that could prove critical toward understanding of disease and development of potential therapies.

Current methods to detect R-loops genome-wide rely on the S9.6 antibody (*Dumelie and Jaffrey, 2017*; *Ginno et al., 2012*; *Nadel et al., 2015*; *Wahba et al., 2016*) or a catalytically inactive RNase H, both of which recognize DNA:RNA hybrids (*Chen et al., 2017*; *Ginno et al., 2012*; *Yan et al., 2019*). DNA–RNA immunoprecipitation-sequencing (DRIP-seq) is the most frequently used S9.6-based approach for R-loop detection genome-wide (*Ginno et al., 2012*). In DRIP, genomic DNA is sheared by enzymatic digestion or sonication and regions that contain R-loops are immunoprecipitated using S9.6 antibody. S9.6-based approaches require high-input material, have low signal-to-noise ratio, and with the exception of BisDRIP-seq (*Dumelie and Jaffrey, 2017*), have

*For correspondence:
kavitha@sarmalab.com

limited resolution. The low signal-to-noise ratio in S9.6 genome-wide experiments may be attributed to the antibody specificity issues that are well documented (*Hartono et al., 2018*; *Vanoosthuyse, 2018*). RNase H methods that include DNA:RNA in vitro enrichment (DRIVE) (*Ginno et al., 2012*), R-loop chromatin immunoprecipitation (R-ChIP) (*Chen et al., 2017*), and MapR (*Yan et al., 2019*) use the evolutionary specificity of the *Escherichia coli* RNase H enzyme that recognizes DNA:RNA hybrids to detect R-loops. While similar to DRIP in the initial steps of sample processing, the in vitro enrichment of R-loops in DRIVE using recombinant catalytically inactive RNase H is inefficient. DRIPc (*Sanz and Chédin, 2019*; *Sanz et al., 2016*) and R-ChIP are both strand-specific techniques with some limitations. DRIPc requires much larger input amounts compared to DRIP and has lower resolution. R-ChIP, a chromatin immunoprecipitation-based strategy, requires the generation of a stable cell line that expresses a catalytic mutant RNase H1 (*Chen et al., 2019*; *Chen et al., 2017*; *Sanz and Chédin, 2019*; *Sanz et al., 2016*) and, while sensitive, may not be amenable for use in all cell types. Therefore, the development of a high-resolution strand-specific R-loop detection strategy that is efficient, sensitive, and amenable to use in all cell types will help in the precise identification of specific regions of the genome that show R-loop anomalies in various diseases.

We recently described MapR (*Yan and Sarma, 2020*; *Yan et al., 2019*), a fast and sensitive R-loop detection strategy founded on the principles of CUT and RUN (*Skene et al., 2018*; *Skene and Henikoff, 2017*) and the specificity of RNase H for the recognition of DNA:RNA hybrids. In MapR, a catalytically inactive RNase H targets microccocal nuclease to R-loops to cleave and release them for high-throughput sequencing. Because MapR is not enrichment-based, unlike DRIP, DRIVE, and R-ChIP, it has high signal-to-noise ratios resulting in enhanced sensitivity (*Yan et al., 2019*). We sought to build on the significant advantages of MapR with respect to specificity, sensitivity, and ease of use and transform it into a strand-specific R-loop profiling strategy. Here, we present BisMapR, a high-resolution, genome-wide methodology that maps strand-specific R-loops.

## Results

### BisMapR identifies strand-specific R-loops

Strand specificity is a key feature of bona fide R-loops. In MapR, cleaved R-loops diffuse out of the nucleus along with mRNAs and other RNAs that are not part of an R-loop. Thus, specific identification of the RNA strand of R-loops as a means of conferring strandedness is challenging with MapR. Instead, we have devised a method to distinguish the template and non-template DNA components of R-loops released by MapR. We leveraged the chemical property of sodium bisulfite to deaminate cytosines (C) to uracils (U) on exposed, single-stranded DNA (ssDNA) only, while double-stranded nucleic acids are protected from conversion (*Gough et al., 1986*). In BisMapR, R-loops released by MapR are treated with sodium bisulfite under non-denaturing conditions (*Figure 1*). This results in the C-to-U conversion of the ssDNA strand of R-loops. Meanwhile, the DNA within the DNA:RNA hybrid is left intact. Next, a second-strand synthesis step replaces the RNA molecule of the DNA:RNA hybrid with a dUTP-containing DNA strand. Adaptors are then ligated to resultant dsDNA. Treatment with uracil DNA glycosylase (UDG) cleaves all dUTP-containing molecules, and the unaltered DNA strand is directionally tagged, PCR amplified, and sequenced. The first-mate reads of the resulting paired-end sequencing data (or all reads for single-end runs) correspond to the DNA:RNA hybrid containing strand and are separated into forward- and reverse-strand tracks.

We compared BisMapR and MapR techniques in mouse embryonic stem cells (mESCs). We distinguished genes as active and inactive based on RNA-seq data. MapR in mESCs specifically identifies R-loops since RNase H catalytic mutant fused to microccocal nuclease (RHΔ-MNase) shows high signal at transcription start sites (TSS) of active genes that are known to form R-loops compared to an MNase-only background control (*Figure 2—figure supplement 1A,B*). MapR signal is also clearly enriched at active genes but not at inactive genes (*Figure 2—figure supplement 1C*). Next, we compared BisMapR to MapR in mESCs to determine whether both techniques showed signal enrichment at similar regions within genes. When examining reads from both strands, which we term 'composite' signal, BisMapR and MapR produce signal enrichment predominantly around the TSS of actively transcribed genes (*Figure 2A*). At the global level also, both datasets showed strong positive correlation for signal enrichment across TSS (*Figure 2—figure supplement 1D*). When reads are separated by originating strand, MapR 'strand-specific' maps are near-identical to their composite

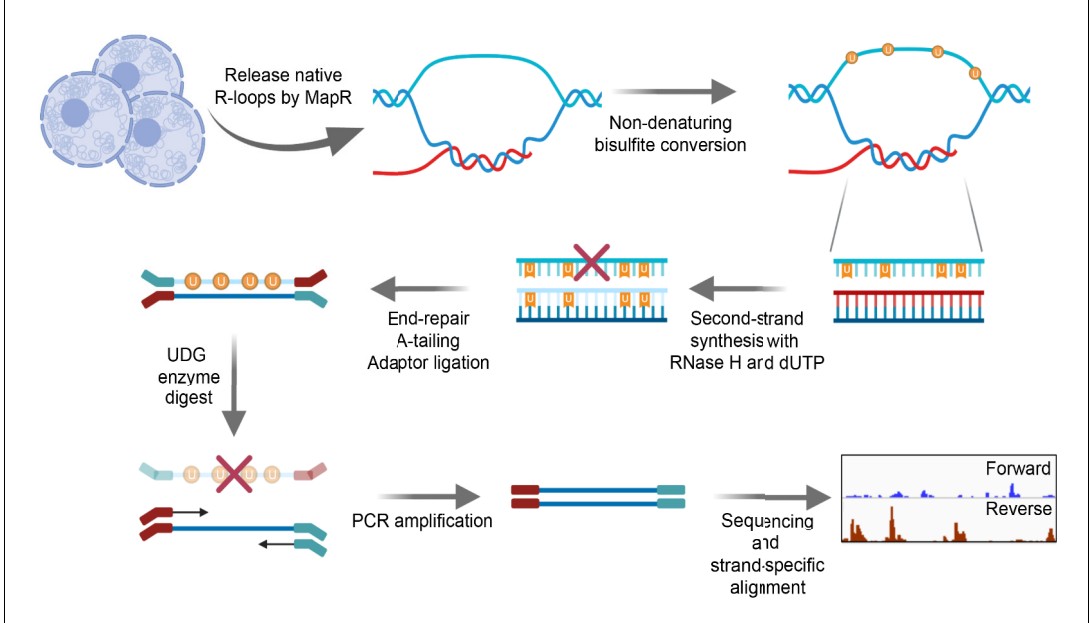

**Figure 1.** BisMapR, an RNase H-based strand-specific native R-loop detection strategy. Schematic of the BisMapR protocol. R-loops are released from cells using MapR and subjected to non-denaturing bisulfite conversion. Bisulfite-converted products are directly processed for second-strand synthesis in the presence of RNase H and dUTP. Adaptors are ligated to resultant dsDNA. Treatment with uracil DNA glycosylase (UDG) degrades all uracil-containing DNA molecules. Remaining DNA is tagged with paired-end barcoded index primers, amplified, sequenced, and strand-specifically aligned. Created with BioRender.com.

(*Figure 2A*), as expected of a non-strand-specific protocol. In contrast, BisMapR reads after strand-specific alignment clearly segregate to either the forward or reverse strands (*Figure 2A*). *Vamp1*, a gene that is transcribed in the sense (plus) direction and expected to produce DNA:RNA hybrids on the reverse (template) strand, shows reverse strand-specific BisMapR signal. Similarly, *Atg10*, an anti-sense (minus) gene, shows BisMapR signal on the forward strand. Global analysis of strand-specific BisMapR signal across all TSS also showed decreased correlation with composite BisMapR as well as with strand-specific and composite MapR (*Figure 2—figure supplement 1E*).

In MapR, released R-loops are treated with RNase A and the resultant dsDNA that forms as a consequence of degradation of the RNA within the DNA:RNA hybrid is processed for library preparation and sequencing using standard dsDNA library preparation protocols (*Yan and Sarma, 2020*). However, non-denaturing bisulfite conversion relies on the presence of ssDNA. To assess whether BisMapR requires intact R-loops to efficiently sort signals by strand of origin, we performed bisulfite conversion reactions after treating MapR samples with RNase A and analyzed forward- or reverse-strand signals across all TSS. BisMapR samples show a clear separation of forward and reverse strands at a large number of TSS in mESCs (*Figure 2B*). We observe that MapR, as a non-strand-specific technique, shows little strand separation (*Figure 2B*) as seen by the almost complete overlap between forward and reverse strand signals. Treatment of R-loops with RNase A prior to bisulfite processing for BisMapR results in a loss of strand specificity that resembles MapR data, with the forward- and reverse-strand signals showing a significant overlap (*Figure 2B*). The residual strand separation in RNase A-treated BisMapR samples is likely due to incomplete RNase A digestion.

Next, we assessed the contribution of the bisulfite conversion step in BisMapR toward achieving strand specificity. For this, we isolated R-loops by MapR, omitted the RNase A digestion step and bisulfite conversion and proceeded to second-strand synthesis directly. R-loop signal from no-bisulfite second-strand (NBSS) samples did not separate well by strand at the vast majority of regions (*Figure 2B*, *Figure 2—figure supplement 1F*), indicating NBSS is not sufficient for strand specificity. The absence of strand specificity in MapR NBSS can occur if R-loop structures released by MapR have intact dsDNA on both ends, which would allow both strands to ligate to adaptors and result in the extruded ssDNA to be incorporated into the sequencing library. However, the MNase digestion

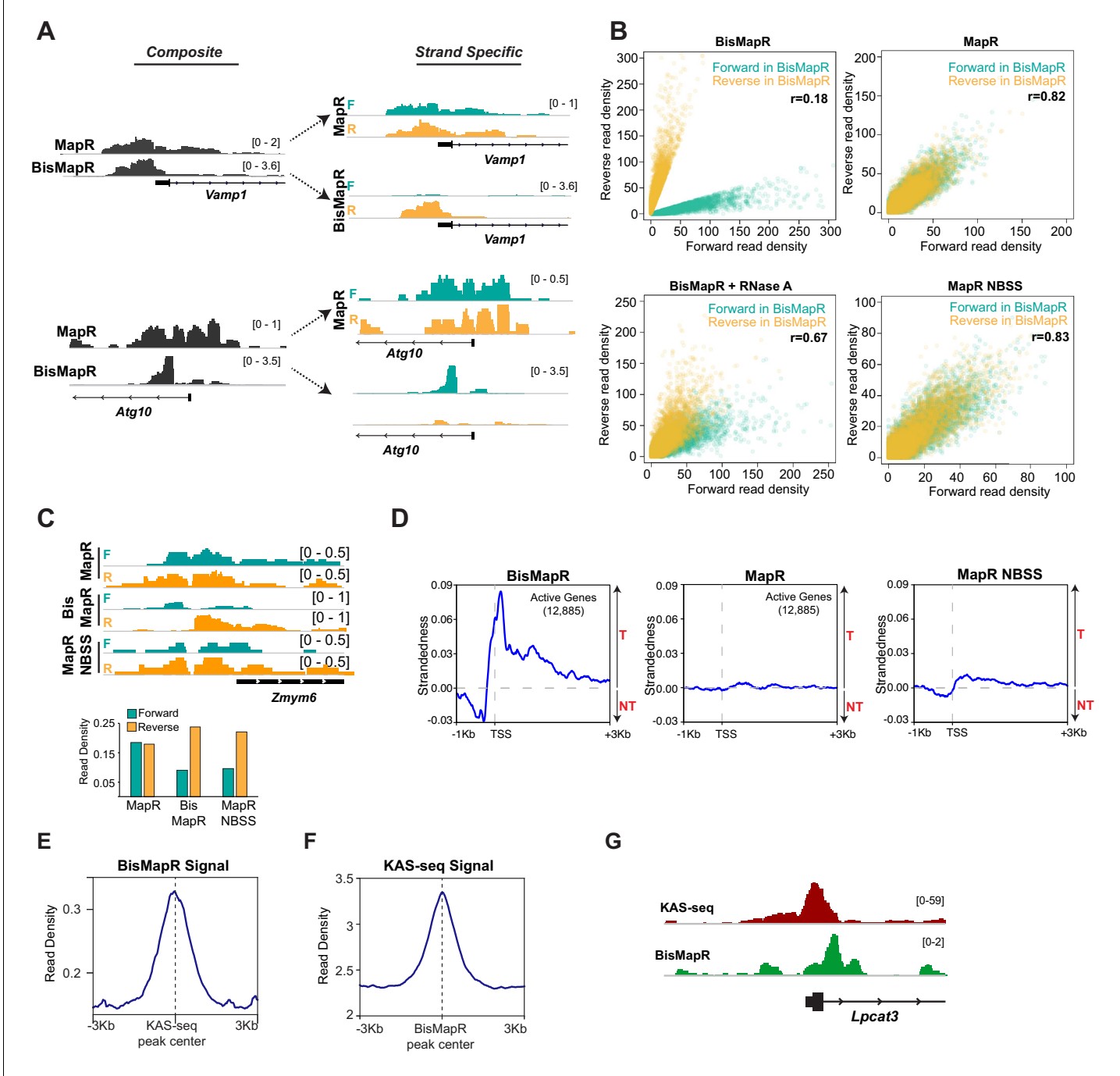

**Figure 2.** BisMapR confers strand specificity to nuclease-based genome-wide R-loop detection. (**A**) Genome browser views of the *Vamp1* and *Atg10* genes showing composite (dark gray) BisMapR and MapR signals (reads per million, RPM) (left) and the same signal when separated into forward (teal) and reverse (orange) strands (right). (**B**) Correlation plots of normalized read densities between forward and reverse strands in BisMapR, MapR, BisMapR samples treated with RNase A, and MapR samples without bisulfite treatment and with second-strand synthesis (NBSS). TSS regions of 19,655 genes with high specificity toward forward (teal, 9788 genes) or reverse (orange, 9867 genes) strand signals in BisMapR, defined as log2 ratio of at least 1.5 in either direction between forward or reverse read densities, are shown. r, Pearson correlation coefficients between forward and reverse read densities for all 19,655 genes. (**C**) Top, genome browser view of the *Zmym6* gene showing forward (teal) and reverse (orange) strand R-loop signal (reads per million, RPM) for MapR, BisMapR, and MapR NBSS. Bottom, bar chart of total forward and reverse strand read densities across the region. (**D**) Strandedness plots of BisMapR, MapR, and MapR NBSS strand-specific signals around the transcription start sites (TSS) of 13,380 active genes in mESCs. Strandedness was calculated as the difference between template (T) and non-template (NT) signal (reads per million, RPM). (**E**) Signal plot of composite BisMapR signal (reads per million, RPM) centered around 40,900 KAS-seq peaks. (**F**) Signal plot of KAS-seq signal centered around 168,774

*Figure 2 continued on next page*

*Figure 2 continued*

composite BisMapR peaks. (**G**) Genome browser view of the *Lpcat3* gene showing KAS-seq (red) and composite BisMapR (green) signal (reads per million, RPM).

The online version of this article includes the following figure supplement(s) for figure 2:

**Figure supplement 1.** BisMapR generates strand-specific transcription-dependent R-loop sequencing data.

step in MapR can also result in the release of DNA:RNA hybrids of the R-loop that are not connected to the ssDNA strand. These DNA:RNA hybrids could contribute to strand-specific signal after second-strand synthesis. The release of ssDNA-free DNA:RNA hybrids is supported by the slightly larger scatter area of data points in MapR NBSS (*Figure 2B*), and by the observation of a small number of regions where NBSS produces strand-specific signal (*Figure 2C*). For example, at the *Zmym6* gene MapR shows near equal R-loop signal from the forward and reverse strands, while BisMapR and MapR NBSS show skewed read distribution with higher signal arising from the reverse (template) strand (*Figure 2C*). Overall, our data show the bisulfite conversion step is critical to ensure removal of the ssDNA component during library preparation and achieve strand specificity.

Divergent transcription is a common feature of active promoters in mammals, with over 75% of active genes producing short antisense RNA transcripts and showing paused RNA polymerase II in the antisense direction (*Core et al., 2008*; *Seila et al., 2008*). This bidirectional transcription implies that in addition to R-loop formation on the template strand associated with transcription of the gene, R-loops would also form on the opposite strand at these divergent promoters. We used Bis-MapR to examine R-loops at promoters of 12,885 active genes in mESCs. As a measure of strand separation in BisMapR, we calculated the 'strandedness' of R-loop signal across active TSS, defined as the difference between template (T) and non-template (NT) signals (*Figure 2D*, left). Upstream of the TSS, strandedness is negative, indicating R-loop signal forms predominantly on the non-template strand consistent with R-loop function in antisense transcription (*Tan-Wong et al., 2019*). Strandedness becomes positive starting around the TSS and continuing into the gene body, as expected from co-transcriptional R-loop formation on the template strand. The clear distinction between template and non-template strand-originating R-loops cannot be made in MapR, where strandedness is entirely absent at any point around TSS (*Figure 2D*, center). Strandedness is only slightly observable in MapR NBSS (*Figure 2D*, right), consistent with only a small subset of reads contributing to strand specificity in NBSS. Thus, our data demonstrate that BisMapR produces strand-specific R-loop profiles genome-wide. We conclude that BisMapR confers strand specificity on intact R-loops that are released by MapR with minor technical modifications and with a minimal burden on time.

R-loops contain DNA:RNA hybrid strand and a single-strand DNA. Thus regions detected by Bis-MapR should also show ssDNA signal. A recently developed technique, kethoxal-assisted single-stranded DNA sequencing (KAS-seq), identifies single-stranded regions of the genome with high G content (*Wu et al., 2020*). In KAS-seq, an azide-tagged kethoxal (*Weng et al., 2020*) that reacts with unpaired guanine residues and that can be modified with a biotin is used to mark and enrich for G-rich regions of the genome that are single stranded. We measured BisMapR composite signal at KAS-seq peaks and found that it is centralized over these regions (*Figure 2E*). KAS-seq signal was also centrally enriched over BisMapR peaks (*Figure 2F*). Consistent with this, KAS-seq and BisMapR signals show high positive correlation (*Figure 2—figure supplement 1G*) and overlap at many active genes (*Figure 2G*). We conclude that BisMapR accurately captures signal at regions of R-loop formation.

## BisMapR can distinguish individual transcriptional units with high resolution

The incorporation of bisulfite treatment that can theoretically react with all single-stranded cytosine residues in the R-loop to mark them for degradation suggests that BisMapR can produce higher resolution R-loop maps compared to MapR. To test this, we analyzed R-loop profiles at active, non-overlapping head-to-head transcriptional units that are separated by less than a kilobase (*Figure 3—figure supplement 1A*). MapR profiles at the 5′ end of *Fbxo18* and *Ankrd16* genes whose TSS are separated by 294 bases show a broad non-strand-specific R-loop enrichment across both TSS (*Figure 3A*). In comparison, BisMapR signal is enriched on the forward strand of *Fbxo18* that is

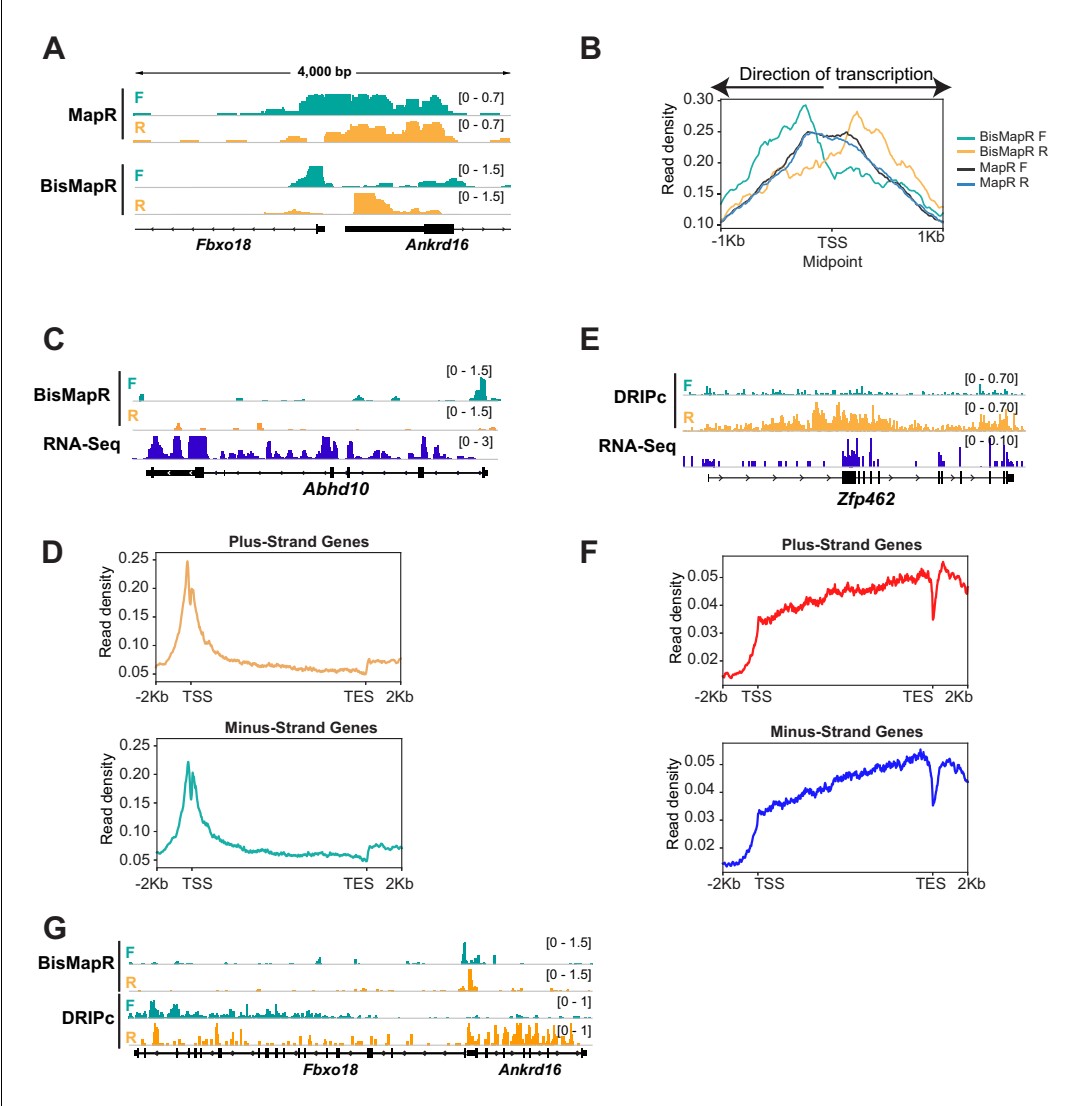

**Figure 3.** BisMapR distinguishes strand-specific R-loops with high resolution at bidirectional gene pairs. (A) Genome browser view of the bidirectional promoter for *Fbxo18* and *Ankrd16* showing MapR and BisMapR signals (RPM) separated by forward (teal) and reverse (orange) strands. (B) Metagene plots of strand-specific BisMapR (forward strand [teal] and reverse strand [orange]) and MapR (forward strand [dark blue] and reverse strand [light blue]) signals at 1001 bidirectional promoters. TSS Midpoint, the midpoint between the transcription start sites of plus-strand and minus-strand genes. (C) Genome browser view of the *Abhd10* gene showing BisMapR signal (reads per million, RPM) separated into forward (teal) and reverse (orange) strands. Mouse ESC RNA-seq signal is shown in blue. (D) Metagene plots of template strand BisMapR signal of 6746 plus-strand genes (top) and 6634 minus-strand genes (bottom) expressed in mESCs. (E) Genome browser view of the *Zfp462* gene showing DRIPc signals separated into forward (teal) and reverse (orange) strands. 3T3 RNA-seq signal is shown in blue. (F) Metagene plots of template strand DRIPc signal of 6223 plus-strand genes (top) and 6164 minus-strand genes (bottom) expressed in 3T3. (G) Genome browser view of the bidirectional promoter for *Fbxo18* and *Ankrd16* showing BisMapR and DRIPc signals (RPM) separated by forward (teal) and reverse (orange) strands.

The online version of this article includes the following figure supplement(s) for figure 3:

**Figure supplement 1.** BisMapR reveals strand specificity of R-loops at high resolution at genic regions genome-wide.

transcribed in the antisense direction and on the reverse strand of *Ankrd16* that is transcribed in the sense direction. A similar profile is observed at the *Zfp146* and *Gm5113* genes whose TSS are separated by 125 bases (*Figure 3—figure supplement 1B*). Global analysis on all bidirectional transcription units demonstrates that BisMapR captures distinct R-loop signals on opposite template strands that are congruent with the direction of transcription (*Figure 3B*). In contrast, MapR signals do not clearly distinguish between pairs of genes (*Figure 3B*). Our data suggests that in addition to

providing strand-specific information, BisMapR improves on the already high resolution of MapR in profiling R-loops genome-wide.

Next, we compared the resolution of BisMapR to DRIPc-seq (*Sanz and Chédin, 2019*; *Sanz et al., 2016*), an S9.6 antibody-based R-loop enrichment method that is most frequently used to identify strand-specific R-loops. In mESCs, we observe that BisMapR signal is highest at the 5′ end of genes and is reduced across the gene body as is seen in the case of *Abhd10* (*Figure 3C*). Metagene analysis across all active genes shows that both plus- and minus-strand genes show 5′ R-loop enrichment (*Figure 3D*). This is consistent with several previous reports that observe R-loop enrichment in proximity to the TSS (*Chen et al., 2017*; *Dumelie and Jaffrey, 2017*; *Ginno et al., 2012*; *Yan et al., 2019*). We used previously published and validated DRIPc data from NIH 3T3 to examine R-loop profiles using this method (*Sanz et al., 2016*). At *Zfp462*, an expressed gene in NIH 3T3 cells, DRIPc signal is broadly present across the entire gene and does not show a clear enrichment at TSS. Surprisingly, analysis of DRIPc signal profiles across all expressed genes on the plus and minus strands shows that DRIPc-seq shows a general enrichment across transcriptional units on the plus and minus strands starting at the TSS and increasing steadily across the gene body (*Figure 3E*).

Our data shows that BisMapR is able to distinguish strand-specific R-loops that form at bidirectional transcription units that are separated by a few hundred bases. To compare the resolution between DRIPc and BisMapR, we visualized R-loops at genes that show head-to-head transcription in both mESC and NIH3T3 cells. At *Fbxo18* and *Ankrd16*, strand-specific BisMapR signal is clearly limited to the TSS of both genes (*Figure 3A and G*). In contrast, DRIPc signal does not show enrichment at TSS and is instead present across the gene body of both genes (*Figure 3G*). Metagene analysis of all active bidirectional transcription units in NIH3T3 shows that DRIPc, while stranded as evidenced by a skew in the forward- and reverse-strand signals in the appropriate direction, does not delineate two distinct transcriptional units (*Figure 3—figure supplement 1C*). Our results indicate that the high resolution of BisMapR, on the order of hundreds of bases, makes it particularly suitable for studying R-loop formation across small-scale features within and between transcriptional units.

## BisMapR reveals unidirectional R-loop formation from KLF7 motifs at a subset of enhancers

Any genomic element with the potential to be transcribed can form R-loops. In addition to genes, enhancers that are transcribed at lower levels and that form short-lived transcripts also form R-loops (*Rabani et al., 2014*; *Schwalb et al., 2016*; *Yan et al., 2019*). To determine differential R-loop formation across enhancers, we examined R-loop signal across active, poised, and primed enhancers in mESCs (*Cruz-Molina et al., 2017*). Active enhancers are bidirectionally transcribed (*Kim et al., 2010*; *Lai et al., 2015*) and show higher global run-on sequencing (GRO-Seq) signals on both the forward and reverse strands as compared to poised and primed enhancers (*Figure 4—figure supplement 1A*). Active, poised, and primed enhancers also contain specific chromatin signatures including histone H3 lysine four monomethylation (H3K4me1) and histone H3 lysine 27 acetylation (H3K27Ac) at active enhancers, H3K4me1 and H3K27 trimethylation (H3K27me3) at poised enhancers, and H3K4me1 at primed enhancers (*Creyghton et al., 2010*; *Rada-Iglesias et al., 2011*; *Zentner et al., 2011*). We found that active enhancers also exhibit higher MapR and BisMapR signals compared to poised and primed enhancers (*Figure 4—figure supplement 1A,B*). Interestingly, clustering of all enhancers based on BisMapR forward- and reverse-strand signals reveals four distinct enhancer clusters: two groups with high R-loops that form on either the reverse (group 1) or forward (group 2) strands, and two groups with medium or low R-loops (groups 3 and 4) (*Figure 4A*, *Figure 4—figure supplement 1C*). Both groups of high R-loop enhancers display R-loops on only one strand. GRO-seq also indicates that group 1 and group 2 enhancers are expressed at higher levels than group 4 enhancers (*Figure 4B*). Group 3 enhancers also had high GRO-Seq levels and may represent enhancers that are expressed without high R-loop formation (*Figure 4B*). Next, we examined the distribution of active, poised, and primed enhancers across these four groups that we defined based on BisMapR profiles. Group 1 is strongly enriched for active enhancers (71%; p=2.49e-167, hypergeometric test for all enrichment p-values) and slightly enriched for poised enhancers (0.05%; p=2.95e-5) (*Figure 4C*). Similarly, group 2 is also enriched for active (72%; p=8.03e-164) and poised enhancers (0.04%; p=0.039). In contrast, the low R-loop

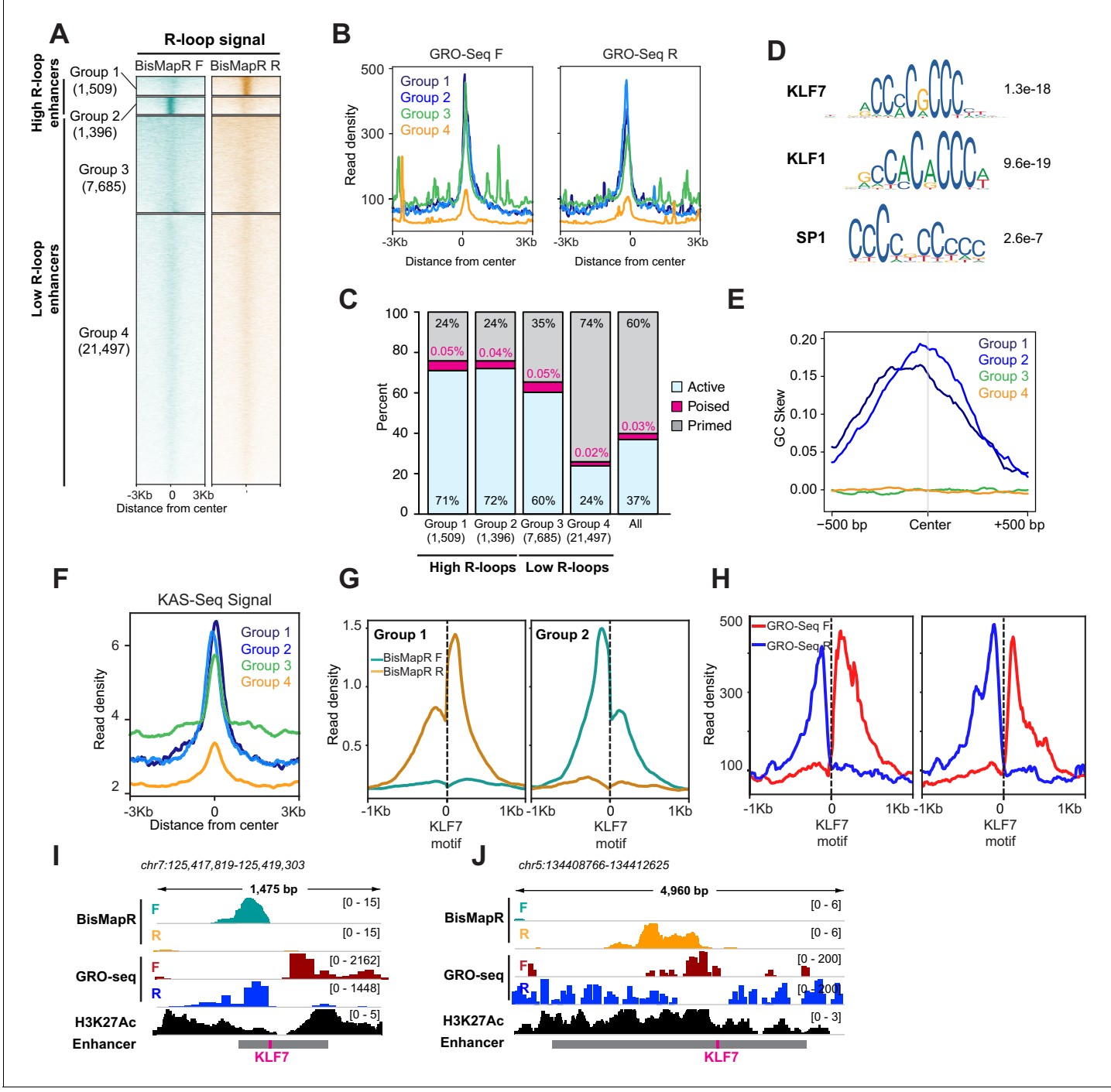

**Figure 4.** BisMapR reveals strand-specific R-loop formation across a subset of enhancers in mESCs. (**A**) Heatmap of strand-specific BisMapR signal across mESC enhancers. Enhancers were divided into four groups by unsupervised k-means clustering (k = 4). Enhancer numbers in each group are indicated in parentheses. Signal, reads per million (RPM). (**B**) Strand-specific GRO-Seq read densities at group 1 (dark blue), group 2 (light blue), group 3 (green), and group 4 (orange) enhancers. (**C**) Barplot showing the proportion of active (blue), poised (pink), or primed (gray) enhancers in each group. Enhancer numbers in each group are indicated in parentheses. Distribution of all enhancer types is shown for comparison. (**D**) Sequence motifs of KLF7, KLF1, and SP1 transcription factors that are centrally enriched across group 1 and 2 enhancers compared to group 3 enhancers. E-values for each sequence are shown. (**E**) GC skew of non-template strand across group 1 (dark blue), group 2 (light blue), group 3 (green), and group 4 (orange) enhancers. GC skew was calculated as (G–C)/(G+C) in 100 bp sliding windows with 10 bp step size. (**F**) KAS-Seq read density across group 1 (dark blue), group 2 (light blue), group 3 (green), and group 4 (orange) enhancers. (**G**) Strand-specific BisMapR signal (RPM) profiles centered around Klf7 motifs identified in group 1 (2094 motifs) and group 2 (1972 motifs) enhancers. (**H**) Strand-specific GRO-seq signal profiles centered around Klf7 motifs identified in group 1 (2094 motifs) and group 2 (1972 motifs) enhancers. (**I, J**) Genome browser view of high-R-loop enhancers showing strand-specific

*Figure 4 continued on next page*

*Figure 4 continued*

BisMapR (F, teal; R, orange) and GRO-seq (F, red; R, blue) signals and H3K27Ac ChIP-seq (black). Enhancer region (gray bar) with location of KLF7 motif (pink) is shown.

The online version of this article includes the following figure supplement(s) for figure 4:

**Figure supplement 1.** BisMapR identifies strand-specific R-loops at enhancers.

(*Figure 4A*) and low transcribed (*Figure 4B*) group 4 is enriched for primed enhancers (74%; p=3.95e-1160) (*Figure 4C*). As expected, genes proximal to group 1 and 2 enhancers, which are enriched for active enhancers, show higher expression levels compared to group 3 and 4 enhancer-related genes (*Figure 4—figure supplement 1D*). To determine whether unidirectional R-loops are observed by other strand-specific R-loop detection strategies, we analyzed DRIPc data from 3T3 cells. Unsupervised clustering of 3T3 enhancers based on DRIPc forward and reverse signals also showed separation into four groups – two high R-loop enhancer groups with DRIPc either on the forward or reverse strands and two low R-loop enhancer clusters (*Figure 4—figure supplement 1E*). Thus, BisMapR uncovers a subset of enhancers that are transcribed in both directions but exhibit unidirectional strand-specific R-loops.

Next, we sought to identify distinguishing features between high and low R-loop enhancers. We used CentriMo to identify sequences that are centrally enriched in the two high R-loop enhancer groups (groups 1 and 2) compared to the low R-loop groups (groups 3 and 4). Our analysis uncovered a significant enrichment for KLF7, KLF1, and SP1 transcription factor motifs in the high R-loop enhancer groups (*Figure 4D*). Since all three motifs appear to have a high GC content, we performed a GC skew analysis to determine whether the non-template strand that forms the ssDNA component of R-loops is enriched for guanines (G). We found that the non-template strands in groups 1 and 2 have high G content, while groups 3 and 4 show no skew (*Figure 4E*). It is possible that the abundance of guanines on the ssDNA can promote the formation of G-quadruplex (GQ) structures that may aid in stabilization of these enhancer R-loops. We compared BisMapR signal at enhancers to KAS-seq signal, which detects ssDNA, and found that high R-loop enhancers as determined by our analysis (groups 1 and 2) exhibit high KAS-seq signal, consistent with our observation of GC skew and the likelihood of containing ssDNA, a component of R-loops (*Figure 4F*). To further determine the degree of concordance between KAS-Seq and BisMapR, we performed clustering of all enhancers based on KAS-seq signal, identifying 5069 enhancers with high KAS-seq signal and 27,811 enhancers with low KAS-seq signal. Seven hundred and four enhancers had both high KAS-seq and high R-loop signals, representing a 1.53-fold enrichment (p=1.96e-35, hypergeometric test). In contrast, enhancers with high R-loops were under-enriched (1.14-fold, p=4.21e-63) in the enhancer population with low KAS-seq signals. Therefore, BisMapR and KAS-seq present two complementary methods to identify enhancers that contain ssDNA components.

Motif analysis at high R-loop enhancers identified an enrichment for KLF7, KLF1, and SP1 binding sites (*Figure 4D*). KLF7, KLF1, and SP1 are pioneer transcription factors that have the ability to bind DNA and open condensed chromatin. Interestingly, analysis of transcription factor binding across DNase I hypersensitive sites showed that KLF7 motif is associated with asymmetrically open chromatin, suggesting a directional pioneer factor activity (*Sherwood et al., 2014*). To examine whether R-loops are skewed at enhancers that contain KLF7 binding sites, we re-centered enhancer regions in groups 1 and 2 around their KLF7 motifs and examined BisMapR signal (*Figure 4G*). Interestingly, in both these groups strand-specific R-loops form unidirectionally from KLF motifs (*Figure 4G, I and J*, *Figure 4—figure supplement 1F*). The presence of H3K27 acetylation (*Figure 4I and J*) and GRO-seq signals (*Figure 4H, I and J*) around the KLF7-centered enhancers show clear bidirectional transcription. Our data suggests that KLF7 binding may contribute to the unidirectional formation or stabilization of R-loops at some active enhancers that show high levels of transcription.

## Discussion

Genome-wide R-loop mapping relies on two distinct approaches, the S9.6 antibody and a catalytically inactive RNase H, that both recognize DNA:RNA hybrids. These two approaches recognize some common R-loops, as well as other unique subsets that likely appear as a consequence of the

distinct sequence preferences of S9.6 and RNase H (*König et al., 2017*). Therefore, orthogonal approaches that are sensitive, that are efficient, and that retain strand specificity will allow for comprehensive interrogation of R-loop dynamics in various cellular processes and their dysfunction in disease. Here, we have described BisMapR, a fast RNase H-based method that efficiently captures strand-specific R-loops at high-resolution genome-wide.

Existing strand-specific R-loop mapping strategies have a few drawbacks that include low-resolution and high-input material (*Sanz and Chédin, 2019*; *Sanz et al., 2016*), involved sample processing (*Dumelie and Jaffrey, 2017*), and lengthy experiment times (*Chen et al., 2019*; *Chen et al., 2017*). Our comparison of BisMapR with DRIPc, an S9.6-based approach, shows that BisMapR produces sharper regions of enrichment suggesting higher resolution (*Figure 3*). The high resolution of BisMapR is especially evident at head-to-head transcribed genes TSS that are only a few hundred bases apart and that show R-loop formation on opposite strands. As noted before (*Sanz and Chédin, 2019*), a reason for the broader signal seen in DRIPc can be because of immunoprecipitation of the regions of the RNA that are not part of the R-loop. While non-denaturing bisulfite treatment can potentially correct this drawback, DRIPc still requires a significantly higher amount of starting material compared to DRIP and MapR. Bisulfite conversion of DRIP samples *after* immunoprecipitation may help reduce input requirement and achieve higher resolution and provide an S9.6-based approach that is on par with the resolution and sensitivity of BisMapR.

Despite differences in R-loop identification between BisMapR and DRIPc, we have previously noted that S9.6- and RNase H-based techniques identify many common R-loops in addition to some that are distinct to each method (*Yan et al., 2019*). For example, while DRIPc does not cleanly separate signal at bidirectional promoters, it still shows skew in the appropriate direction at these regions (*Figure 3—figure supplement 1C*). DRIPc also identifies enhancers with broadly increased R-loop signal on one strand only (*Figure 4—figure supplement 1D*). The differences between S9.6- and RNase H-based methods could arise from the molecular preferences of S9.6 antibody and RNase H. In vitro studies show that S9.6 prefers binding R-loops with lower GC content (*König et al., 2017*). Whether RNase H preferences to a specific R-loop structure is not yet known.

At active gene promoters, BisMapR identifies both non-template signal upstream of the TSS that is consistent with antisense divergent transcription, and template signal centered around the TSS and continuing downstream into the gene body. Interestingly, template signal is not confined strictly downstream of the TSS, but instead is enriched slightly upstream and predominantly downstream of the TSS with a slight dip in signal over the TSS itself. This is a pattern also observed in MapR (*Figure 2—figure supplement 1B*). MapR relies on the docking of the GST-tagged catalytically inactive RNaseH (RHΔ) on R-loops and the subsequent cleavage around the interaction site by the MNase moiety. We predict that the reason we see a broad signal that extends upstream of TSS is because of where R-loops start, the binding of GST-RHΔ-MNase protein at these sites and the ability of MNase moiety to access the upstream region when bound. These together would result in the appearance of signal slightly upstream and downstream of the TSS. Another possibility is that BisMapR can detect R-loops forming from alternative TSS usage, or extra-coding RNAs (ecRNAs) that precede TSS, both of which would result in signal upstream of annotated start sites.

Using BisMapR, we uncovered a subset of enhancers with high R-loops and that have a high GC skew. R-loop formation is associated with the potential to form G quadruplexes on the non-template strand. Two recent studies using single-molecule approaches have provided insight into how G quadruplexes and R-loop formation regulate gene expression. While R-loop formation precedes GQ formation, stable GQs in the non-template strand provide a positive feedback to promote R-loops during transcription (*Lim and Hohng, 2020*). Transcription efficiency is increased as a result of successive rounds of R-loop formation (*Lee et al., 2020*). Taken together with our finding of high R-loops at some enhancers, it is possible that R-loop formation and GQ stabilization lead to the maintenance of nucleosome-free regions and help in sustained enhancer activation.

In summary, BisMapR is a fast, sensitive, and strand-specific R-loop detection strategy that reveals strand-specific R-loops at enhancers that are also enriched for KLF7 pioneer factor-binding motifs. Our study provides a tool to further dissect how directional chromatin accessibility conferred by a subset of pioneer factors contributes to R-loop stabilization at enhancers and their combined significance to gene expression.

## Materials and methods

### Cell culture

E14 mouse embryonic stem cells (mESCs) were a gift from Dr. Roberto Bonasio's lab at the University of Pennsylvania. Mouse ESCs were cultured on 0.1% gelatin-coated plates in media containing DMEM, 15% fetal bovine serum (Gibco), 1× MEM non-essential amino acids, 1× GlutaMAX (Gibco 35050), 25 mM HEPES, 100 U/ml Pen-Strep, 55 µM 2-mercaptoethanol, 3 µM glycogen synthase kinase inhibitor (Millipore 361559), 1 µM MEK1/2 inhibitor (Millipore 444966), and LIF (Sigma, ESGRO). The pluripotent state of these cells was ascertained by expression analysis of markers including Oct4 and Nanog by RNA sequencing. Cell lines tested negative for mycoplasma.

### BisMapR

MapR was performed as described (*Yan and Sarma, 2020*; *Yan et al., 2019*) on $5 \times 10^6$ cells for Bis-MapR and control MapR samples (n = 2 replicates per method), with the exception that RNase A was omitted from the stop buffer of the BisMapR sample. Following DNA extraction, the BisMapR samples were bisulfite converted using reagents from the EZ DNA Methylation-Gold Kit (Zymo D5005). Ten microliters of sample was added to 10 µL of $dH_2O$ and 130 µL of CT Conversion Reagent, then incubated for 3 hr at room temperature (25°C) to preserve double-stranded nucleic acids under non-denaturing conditions. DNA desulfonation and column purification were performed according to manufacturer's instructions and eluted into 20 µL of M-Elution buffer. The elution product was directly used for second-strand synthesis using reagents from the NEBNext Ultra II Directional RNA Library Prep Kit (NEB E7760). Eight microliters of Second-Strand Synthesis Reaction Buffer, 4 µL Second-Strand Synthesis Enzyme Mix, and 48 µL $dH_2O$ was added to elution product, and the reaction was incubated for 1 hr at 16°C in a thermocycler. Double-stranded DNA was purified from the second-strand reaction using 1.8× volume of AMPure XP SPRI beads (Beckman Coulter). As a negative control, BisMapR bisulfite conversion and second-strand synthesis steps were also performed on a MapR sample in which RNase A was present in the stop buffer (BisMapR + RNase A). A MapR sample without RNase A was also subjected to the second-strand synthesis step without bisulfite conversion (MapR NBSS).

### RNA-Seq

RNA samples were extracted from mESCs (n = 3 biological replicates) using Trizol reagent (Invitrogen) and subjected to DNase digestion with Turbo DNase (Ambion AM2238). RNA samples were then rRNA depleted using FastSelect -rRNA HMR (Qiagen) and converted to cDNA using Ultra II Directional RNA Library Prep Kit (NEB E7760).

### Library preparation and sequencing

DNA samples were end-repaired using End-Repair Mix (Enzymatics), A-tailed using Klenow exonuclease minus (Enzymatics), purified with MinElute columns (Qiagen), and ligated to Illumina adaptors (NEB E7600) with T4 DNA ligase (Enzymatics). Size selection for fragments > 150 bp was performed with AMPure XP (Beckman Coulter). Libraries were PCR amplified with dual-index barcode primers for Illumina sequencing (NEB E7600) using Q5 DNA polymerase (NEB M0491) and purified with MinElute. Uracil DNA glycosylase (Enzymatics) was added to the PCR amplification mix to degrade dUTP-containing molecules and remove adaptor hairpins. Sequencing was performed on a NextSeq 500 instrument (Illumina) with 38 × 2 paired-end cycles.

### Data processing

BisMapR, MapR, and DRIPc-seq reads were mapped to the mouse genome (mm10) with Bowtie2 version 2.2.9 (*Langmead and Salzberg, 2012*) using default parameters and paired-end (BisMapR and MapR) or single-end (DRIPc-seq) settings as appropriate. Mouse ESC and NIH3T3 RNA-Seq reads were mapped to mm10 with STAR version 2.7.3 (*Dobin et al., 2013*), and RSEM (*Li and Dewey, 2011*) version 1.3.3 was used to obtain estimated counts. A gene was considered expressed if it had at least one count per million in all RNA-Seq samples. A bidirectional promoter was defined as a region 1 kb or smaller containing two transcription start sites for genes in opposite directions, that is sense and antisense, and with both genes expressed in mESC or NIH3T3. To generate strand-

specific datasets for BisMapR, MapR, RNA-Seq, and DRIPc-seq, reads were separated based on the strand to which the first mate aligned. Specifically, reads with SAM flags 16, 83, or 163 were placed into the forward-strand dataset, while reads with SAM flags 0, 99, or 147 were placed into the reverse-strand dataset. The first-mate strand represents the template strand for BisMapR, MapR, and DRIPc-seq. RPM normalization for strand-specific datasets was calculated based on the combined number of reads assigned to either the forward or reverse strand. BigWig tracks were generated using the bamCoverage function in deepTools 3.4.1 (*Ramírez et al., 2016*) with options `–binSize` five and `–blackListFileName` to remove a known set of ENCODE blacklist regions (*Amemiya et al., 2019*). The `–extendReads` option was used for paired-end datasets. Strandedness was calculated by subtracting reverse- strand BigWig signal from forward-strand BigWig signal for reverse-strand genes and vice versa for forward-strand genes. Peak calling of BisMapR composite samples was performed using MACS2 version 2.1.1 (*Zhang et al., 2008*). Signal plots and heatmaps were generated using the computeMatrix, plotProfile, and plotHeatmap functions in deepTools. Motif enrichment analysis was performed with CentriMo using 500 bp sequences centered around each enhancer.

## Published data

We downloaded FASTQ files for NIH3T3 DRIPc-seq (SRR3322169) and NIH3T3 RNA-seq (SRR6126847), BigWig tracks and peak locations for KAS-seq (*Wu et al., 2020*) (GSE139420), GRO-seq (*Tastemel et al., 2017*) (GSE99760) and H3K27Ac (ENCODE ENCFF163HEV), mESC enhancer locations from *Cruz-Molina et al., 2017*, and 3T3 enhancers from the Enhancer Atlas (*Gao and Qian, 2020*).

## Acknowledgements

We thank A Gardini and R Bonasio for helpful discussions, and Q Yan for assistance with MapR experiments. This work was supported by National Institutes of Health Grants DP2-NS105576 (to KS) and T32CA009171 (to PW).

## Additional information

### Competing interests

Kavitha Sarma: Reviewing editor, *eLife*. The other author declares that no competing interests exist.

### Funding

| Funder | Grant reference number | Author |
|---|---|---|
| NIH Office of the Director | DP2-NS105576 | Kavitha Sarma |
| National Institutes of Health | T32CA009171 | Phillip Wulfridge |

The funders had no role in study design, data collection and interpretation, or the decision to submit the work for publication.

### Author contributions

Phillip Wulfridge, Conceptualization, Software, Formal analysis, Investigation, Writing - original draft, Writing - review and editing; Kavitha Sarma, Conceptualization, Supervision, Writing - review and editing

### Author ORCIDs

Phillip Wulfridge (iD) https://orcid.org/0000-0002-7195-6814
Kavitha Sarma (iD) https://orcid.org/0000-0002-3045-210X

### Decision letter and Author response

Decision letter https://doi.org/10.7554/eLife.65146.sa1
Author response https://doi.org/10.7554/eLife.65146.sa2

## Additional files

### Supplementary files
• Transparent reporting form

### Data availability

Sequencing data generated for this study have been deposited in the NCBI GEO as GSE160578.

The following dataset was generated:

| Author(s) | Year | Dataset title | Dataset URL | Database and Identifier |
|---|---|---|---|---|
| Wulfridge P, Sarma K | 2020 | BisMapR: a strand-specific, nuclease-based method for genome-wide R-loop detection | https://www.ncbi.nlm.nih.gov/geo/query/acc.cgi?acc=GSE160578 | NCBI Gene Expression Omnibus, GSE160578 |

The following previously published datasets were used:

| Author(s) | Year | Dataset title | Dataset URL | Database and Identifier |
|---|---|---|---|---|
| Lyu R, Wu T | 2020 | In situ capture of global transcription dynamics and enhancer activity with high accuracy and sensitivity | https://www.ncbi.nlm.nih.gov/geo/query/acc.cgi?acc=GSE139420 | NCBI Gene Expression Omnibus, GSE139420 |
| Xiaoying B | 2017 | Transcription Pausing Regulates Mouse Embryonic Stem Cell Differentiation | https://www.ncbi.nlm.nih.gov/geo/query/acc.cgi?acc=GSE99760 | NCBI Gene Expression Omnibus, GSE99760 |

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
