## [Decision Letter]

**Acceptance summary:**

The authors provide an improved method to detect R-loops genome-wide. The method provides improved signal-to-noise ratio and strand-specific information that are advantageous to existing methods. This method should be useful with increasing interest in R-loops in control of gene regulation and genome stability.

**Decision letter after peer review:**

Thank you for submitting your article "BisMapR: a strand-specific, nuclease-based method for genome-wide R-loop detection" for consideration by *eLife*. Your article has been reviewed by three peer reviewers, including Howard Y Chang as the Reviewing Editor and Reviewer #1, and the evaluation has been overseen by Kevin Struhl as the Senior Editor.

The reviewers have discussed the reviews with one another and the Reviewing Editor has drafted this decision to help you prepare a revised submission.

Summary:

Wulfridge and Sarma describe a new approach to map R-loops genome-wide. Using sodium bisulfite conversion, the ssDNA of the R-loop is targeted and specifically the DNA hybridized to the RNA in the R loop is sequenced. The approach yields strand specific information on where R-loops form and at high resolution. While the approach might be an important advance, some results don't make sense which prompts concerns about the possibility of artifacts or strong biases.

Essential revisions:

1) Comparison of BisMapR with alternative methods to clarify the methodologic advance.

a) If strand-specific library construction kit is used in MapR the authors may achieve strand-specific detection already. Please justify why bisulfite treatment is needed.

b) The authors compare their data to another approach that maps R-loops in a strand specific manner, DRIPc-seq. The two methods give strikingly different results. The authors do not give any rationale for why this is. Could BisMapR be extremely selective? Perhaps it is only mapping the most stable R-loops, those that can form G-quadraplexes or are stabilized by another mechanism.

c) Comparison with single stranded DNA detection method like KAS-seq would be informative. Single-stranded DNA is a kind of proxy for R-loops. BisMap-R should provide richer data but may need more cells and may not be applicable to clinical tissue samples.

2) Biological insight of divergent transcription at promoters and enhancers.

Divergent transcription: The authors rightfully cite the bias of R-loops on the template strand as evidence that their approach works (Figure 2C). They also mention that they see R-loops in the region upstream of TSSs, which they reason arrive from antisense divergent transcription seen at upstream of most mammalian promoters. However, the authors see both template and non-template signal at these locations at identical levels. How is this possible? There wouldn't be template RNA before the TSS. With the purported resolution of the method, this should be resolvable.

Enhancers: The authors find that bidirectionally transcribed enhancers have an R-loop only on one strand. This is extremely striking and if true, a remarkable discovery. However, it may just reflect the propensity for BisMapR to map the most stable R-loops.

a) How does DRIPc-seq data look like at these enhancers?

b) The fact that some features are so different between group 1 and group 2 enhancers is confusing. First there are so many fewer Group 2 enhancers compared to Group 1. It suggests an asymmetry to which strand of the DNA R-loops form in, which does not seem possible. And then there is a higher GC skew for group 2 enhancers. The strand asymmetries are either clues as to what is going on or a potential computational artifact.

---

## [Author Response]

Essential revisions:1) Comparison of BisMapR with alternative methods to clarify the methodologic advance.a) If strand-specific library construction kit is used in MapR the authors may achieve strand-specific detection already. Please justify why bisulfite treatment is needed.

We thank the reviewers for this suggestion. To determine if we could achieve strand-specific detection with MapR, we omitted RNase A at the R-loop collection step and directly proceeded to second strand synthesis (MapR no bisulfite second strand – NBSS). Our results indicate that MapR NBSS sample are **not** strand-specific (Figure 2B); instead, reads are distributed almost evenly across both strands like the MapR sample. However, we detect a slight skew in the MapR NBSS reads obtained congruent with the direction of transcription (Figure 2D, E, and Figure 2—figure supplement 1F). This result indicates that second-strand synthesis alone is insufficient for removing the ssDNA component of R-loops during library preparation and that bisulfite treatment is need to confer strandedness to MapR samples.

One explanation for our observed results is that intact R-loops cleaved and released by MapR may be flanked by dsDNA – in fact, this is highly likely considering that the original MapR protocol (Yan et al., 2019), which adds RNase A to digest RNA, generates dsDNA molecules immediately suitable for library preparation. Such intact R-loops could be successfully adaptor-ligated on both strands despite partial ssDNA within the structure. Both strands would then be amplified and sequenced. With this rationale, RNA:DNA hybrids that are separated from ssDNA during MNase cleavage – that is, molecules without dsDNA ends – would be successfully converted and sequenced in a strand-specific manner by both BisMapR and MapR NBSS. Indeed, we identify a subset of R-loops where MapR NBSS signal is skewed and resembles BisMapR signal (Figure 2E). Notably, many of these regions have low MapR signal, consistent with RNA:DNA hybrid molecules at these regions being converted to ssDNA by addition of RNase A and subsequently lost in MapR libraries. Our analyses indicate that both bisulfite conversion and second-strand synthesis are instrumental to fully capturing R-loops in a strand-specific manner.

These results are detailed in the revised manuscript.

b) The authors compare their data to another approach that maps R-loops in a strand specific manner, DRIPc-seq. The two methods give strikingly different results. The authors do not give any rationale for why this is. Could BisMapR be extremely selective? Perhaps it is only mapping the most stable R-loops, those that can form G-quadraplexes or are stabilized by another mechanism.

We think the main reason for differences obtained upon comparison of BisMapR and DRIPc (Figure 2) is the resolution of each technique. BisMapR offers higher resolution that DRIPc. As noted in a recent protocol paper by the authors who developed DRIPc (Sanz and Chedin, 2019), a potential limitation of DRIPc, that can result in false positive signal and decrease resolution, is the presence of “trailing” peaks that arise from the free portion of the RNA that is not involved in R-loop formation. However, our analysis of bi-directional promoters using BisMapR and DRIPc do not show contradicting results. While DRIPc does not clearly separate bidirectional promoters, it still shows a clear skew in the appropriate direction at these promoters (Figure 3—figure supplement 1C). Our enhancer analyses also show agreement between BisMapR and DRIPc (Figure 4). Both BisMapR and DRIPc discover unidirectional R-loops at a subset of active enhancers in mESCs and 3T3, respectively (Figure 4A, Figure 4—figure supplement 1E-F).

Additionally, we have shown previously that S9.6 and RNase H based techniques identify many common R-loops and also some that are distinct (Yan et al., 2019). DRIPc uses S9.6 to capture DNA:RNA hybrids. in vitro studies show that S9.6 prefers binding R-loops with lower GC content (Konig, Schubert and Langst, 2017). Whether RNase H has molecular preferences to a specific R-loop structures is not known. We have included this in the Discussion in the revised manuscript.

c) Comparison with single stranded DNA detection method like KAS-seq would be informative. Single-stranded DNA is a kind of proxy for R-loops. BisMap-R should provide richer data but may need more cells and may not be applicable to clinical tissue samples.

We had utilized KAS-seq data in our analysis of enhancer elements, and now perform a more global comparison of BisMapR to KAS-Seq. We examined BisMapR signal over KAS-seq peaks obtained from literature and found that BisMapR was centrally enriched across these KAS-seq peaks, indicating that BisMapR captures signal over ssDNA regions that correspond to R-loops (Figure 2G). In our analyses, KAS-seq signal was also centrally enriched over BisMapR peaks (Figure 2H). The two datasets show strong positive correlation (Pearson correlation coefficient = 0.99). Consistent with this global finding, KAS-seq and BisMapR signals show enrichment at many shared genomic sites. We have added this analysis to the manuscript.

KAS-seq and BisMapR do not completely overlap. Sites where KAS-seq signal is present, but not BisMapR, may represent single stranded areas of the genome that are R-loop independent (Kouzine, Cell Syst. 2017). Similarly, regions with BisMapR, but not KAS-seq may either be a result for incomplete kethoxal reactivity or dynamic R-loops that are only captured by BisMapR. However, because these require further experimental proof to elucidate the true structure of nucleic acids at these regions, we have not discussed these in the manuscript.

2) Biological insight of divergent transcription at promoters and enhancers.Divergent transcription: The authors rightfully cite the bias of R-loops on the template strand as evidence that their approach works (Figure 2C). They also mention that they see R-loops in the region upstream of TSSs, which they reason arrive from antisense divergent transcription seen at upstream of most mammalian promoters. However, the authors see both template and non-template signal at these locations at identical levels. How is this possible? There wouldn't be template RNA before the TSS. With the purported resolution of the method, this should be resolvable.

We thank the reviewers for this insightful observation. We have performed a more detailed breakdown of R-loop distribution at active promoters in order to bring more clarity to this result.

First, we think the confusion regarding template and non-template signal appearing equivalent upstream of the TSS arises from the nature of how metagene plots combine signal. When individually examined, upstream R-loop signal can be broadly classified into two types. In a relatively small subset of active genes, strong non-template R-loop is strictly confined upstream of the TSS. In contrast, a larger group of active genes has template R-loop signal centered around the TSS; this signal begins slightly upstream of the TSS and continues into the gene body. Notably, non-template and template signal is generally mutually exclusive, indicating that BisMapR achieves strand separation at these upstream regions. However, because there are fewer genes with strong non-template signal upstream of TSS than genes with template signal around the TSS, averaging signal across all active TSS in a metagene plot gives the false impression that template and non-template strands are equal. To avoid this confusion, we have replaced the metaplot with a measure of “strandedness” obtained by subtracting non-template from template signal at each individual region (Figure 2F). This computation more accurately reflects the predominance of non-template signal upstream of the TSS, followed by template signal around and past the TSS.

Regarding the presence of template strand upstream of the TSS, we have consistently observed that template and non-template strands show a “two-peak” pattern. This is most apparent on active genes with template strand signal, where a broad peak appears to have a dip at the TSS and signal extending slightly upstream and further downstream. Interestingly, this is a pattern we observe at the same genes across several independent MapR/BisMapR experiments in mESCs (Figure 2C and Figure 2—figure supplement 1B). We notice that upstream of the TSS, this signal drops off around the 150bp mark, while it decreases more gradually downstream of the TSS. MapR relies on the docking of the GST tagged catalytically inactive RNaseH (RH∆) on R-loops and the subsequent cleavage around the interaction site by the MNase moiety. We predict that the reason we see signal extending upstream of TSS is because of where R-loops start, the binding of GST-RH∆-MNase protein at these sites and the ability of MNase moiety to access the upstream region when bound. This results in the appearance of signal that originates from the template RNA upstream of TSS. Alternatively, the actual transcription start site for some genes might be upstream of the annotated position we use either because of alternative TSS usage or because of the existence of upstream “extra-coding” RNAs (ecRNAs) (Di Ruscio et al. Nature 2013). We now include discussion of these possibilities in the revised manuscript.

Enhancers: The authors find that bidirectionally transcribed enhancers have an R-loop only on one strand. This is extremely striking and if true, a remarkable discovery. However, it may just reflect the propensity for BisMapR to map the most stable R-loops.a) How does DRIPc-seq data look like at these enhancers?

Because DRIPc data from mESC is not available, we analysed DRIPc at NIH 3T3 enhancers. We obtained a list of enhancers in NIH-3T3 cells from the Enhancer Atlas (http://www.enhanceratlas.org/) and analyzed NIH-3T3 DRIPc-seq signal over these regions. Mirroring our findings in E14 mESCs, a subset of 3T3 enhancers contain DRIPc-seq signal on either the forward or reverse strand only (Figure 4—figure supplement 1E-F). However, because of the resolution limit of DRIPc, the strand specific signal across enhancers is more diffuse in comparison to BisMapR’s narrower and highly centralized signal. Thus, both BisMapR and DRIPc reveal unidirectional R-loops at enhancers. This is now detailed in the revised manuscript.

b) The fact that some features are so different between group 1 and group 2 enhancers is confusing. First there are so many fewer Group 2 enhancers compared to Group 1. It suggests an asymmetry to which strand of the DNA R-loops form in, which does not seem possible. And then there is a higher GC skew for group 2 enhancers. The strand asymmetries are either clues as to what is going on or a potential computational artifact.

We thank the reviewers for pointing out this potential source of confusion. They are correct in that the ostensible imbalance in number and GC skew between Group 1 and Group 2 enhancer stems from a computational step, namely the unsupervised k-means clustering step. In this case, the clustering algorithm grouped some regions of medium R-loop formation into the reverse-R-loop cluster (Group 1). While we do not believe this grouping affects our overall conclusions, we agree that using a method that reduces the asymmetry between groups will help reduce confusion and increase the clarity of our results.

We have thus used the k-means algorithm to group enhancers into four clusters, rather than three, and found that this resolves the issue: the two high R-loop groups have universally strong R-loop signal on either the forward or reverse strand only (Figure 4A) , are more equivalent in the number of enhancers in each group, and have comparable GC skews (Figure 4E). High R-loop groups remain centrally and significantly enriched for KLF7, KLF1, and SP1 motifs (Figure 4D), and our overall observations on the nature of these unidirectional R-loop enhancers remain unchanged. We have updated Figure 4 and Figure 4—figure supplement 1C to reflect our new 4-cluster analysis and edited text and enhancer numbers as appropriate.